environmental science

compactness, hotspot,
urban economic growth

**Author for correspondence:**
Xun Li
e-mail: lixun@mail.sysu.edu.cn

# The inverted U-shaped effect of urban hotspots spatial compactness on urban economic growth

Weipan Xu[1], Haohui Chen[2], Enrique Frias-Martinez[3], Manuel Cebrian[4,5] and Xun Li[1]

[1]Sun Yat-sen University, Guangzhou, China
[2]Commonwealth Scientific and Industrial Research Organisation, Canberra, Australian Capital Territory, Australia
[3]Telefonica Research, Madrid, Spain
[4]Massachusetts Institute of Technology Media Lab, Cambridge, MA, USA
[5]Center for Humans and Machines, Max Planck Institute for Human Development, Berlin, Germany

WX, 0000-0003-1614-3957; HC, 0000-0001-8976-3634;
XL, 0000-0002-7190-0853

The compact city, as a sustainable concept, is intended to augment the efficiency of urban function. However, previous studies have concentrated more on morphology than on structure. The present study focuses on urban structural elements, i.e. urban hotspots consisting of high-density and high-intensity socioeconomic zones, and explores the economic performance associated with their spatial structure. We use night-time luminosity data and the *Loubar* method to identify and extract the hotspot and ultimately draw two conclusions. First, with population increasing, the hotspot number scales sublinearly with an exponent of approximately 0.50–0.55, regardless of the location in China, the EU or the USA, while the intersect values are totally different, which is mainly due to different economic developmental level. Secondly, we demonstrate that the compactness of hotspots imposes an inverted U-shaped influence on economic growth, which implies that an optimal compactness coefficient does exist. These findings are helpful for urban planning.

## 1. Introduction

Urban sprawl has been an area of active research over the past decades. In Western countries, some studies suggest that urban sprawl has led to environmental deterioration and social problems [1–3]. China, at its present stage of rapid urbanization,

is also undergoing severe urban expansion which has resulted in the emergence of 'ghost towns' [4]. Scholars have presented a sustainable concept, the 'compact city', to address these adverse effects. This concept has attracted attention, particularly after the Commission of the European Communities (CEC) advocated the compact city concept in 1990 as an approach to solve housing and environmental problems [5]. Such cities are characterized by high-density and multifunctional land use and a compact urban form [6,7]. However, as opponents point out, over-compactness can place great pressure on the inner-city environment [8], leading to high house prices and social deterioration [9]. Although the concept of sustainability represents a common and fundamental goal, the appropriate compactness of an urban area is still contentious. The optimal level of compactness to ensure satisfactory city performance requires further discussion.

Compactness is related not only to urban density but also to structure, specifically, the spatial arrangement of urban hubs and centres. The research on urban structure originated from the Alonso–Mills–Muth monocentric model [10–13] and Krugman's core-peripheries urban model [14]. Both models attribute the agglomeration of population to the economy of scales and transportation costs for goods, and theoretically explain the forces driving the regional transition from isolated small settlements to a concentrated core urban area. However, given that the transportation costs for goods has been dramatically declining since 1960 [15], dispersion should have been dominating, resulting in vanishing agglomerations and limitless sprawls in cities [16]. Multiple studies have demonstrated that the urban spatial structure tends to be polycentric when the population size increases, specifically with the declining transportation cost [17–19].

The economic influence of urban structures has attracted much attention. Some researchers found polycentric spatial structures can promote urban economic growth [20,21], while others have found that spatial structures actually have no effect on population or employment growth [22]. The degree of polycentrism is typically based on the degree of the rank-size distribution [20,23,24]. When measuring city compactness, scholars often focus on density, mixed-land use and urban form but rarely consider the structural elements. Considering that networks and nodes allow urban components, such as people and goods, to interplay [25], it is relevant to study the economic performance of compact cities from the perspective of its structural components.

Urban hotspots (or activity centres), the most crowded places with social and economic activities, can be considered as the structural components of an urban environment. Urban performance depends on the appropriate proportion of density and links, so that every part is integrated into a region and that regions are integrated into the complete city [26]. Since a city constitutes a complete integration, hotspots must interact with one another to maintain the function of the urban network. For the identification of hotspots, population is often gridded with different resolutions, based on which researchers build algorithms to identify urban clusters [27,28]. The classic method to identify hotspots is to find out the threshold of population density, as exemplified by the work done in Los Angeles [29]. However, the choice of threshold is arbitrary and disputable. Louail *et al.* [27] have proposed a non-parametric method based on the Lorenz curve, which can generate a threshold *endogenously* according to the density distribution.

This paper uses night-time luminosity (NTL) data and the Lorenz curve method to study how hotspots change with population, how hotspots are spatially organized and how the compactness of hotspots affects urban economic growth, using a global combination of US, EU and Chinese cities.

# 2. Material and methods

## 2.1. Study cases

Our study uses a sample of cities from China, the USA and the EU to study the effect of hotspots in economic development. Cities in China refer to municipal districts, as defined by the Database of Global Administrative Areas (http://www.gadm.org/). The economic statistical data regarding population and GDP originates from the *Six National Population Census, China City Statistical Year Book.* Cities in the EU refer to Territorial Units for Statistics (NUTS), a geocode standard developed by the EU for referencing the subdivisions of countries for statistical purposes (http://ec.europa.eu/eurostat/data/). As for the USA, cities refer to Metropolitan Statistical Areas, as defined by the Bureau of Economic Affairs (https://www.bea.gov/, https://www.census.gov/). The number of urban areas considered in the USA, the EU and China are, respectively, 349, 239 and 276. Figure 1 presents all cities used in this study, including their definition and the hotspots identified.

**Figure 1.** Location of cities used in this study across (*a*) the USA, (*b*) the EU and (*c*) China. The number of urban areas considered are 349, 239 and 276, respectively. Hotspots are shown inside the city boundary.

## 2.2. Night-time luminosity data

Understanding urban environments involves the collection and analysis of high-resolution socioeconomic data [20,30] or telecommunication data [27], both of which are costly to acquire and sometimes lack timely updates. In particular, in developing economies, such datasets may be sampled too coarsely or are unavailable to the public. Night-time luminosity (NTL) data, which do not have the previous limitations, have been used to explore urban economic activities, such as mapping the urban transition [31], measuring urban growth [32,33] analysing urban spread [34] and analysing the spatial heterogeneity of human activities within a city [35].

In this study, we use global-radiance-calibrated night-time light (GRCNL) collected by the Defence Meteorological Satellite Program's Operational Linescan System (DMSP-OLS), which has a spatial resolution of 30 arc-seconds (approx. 1000 m of earth surface) as provided by the Nation Geophysical Data Center (https://ngdc.noaa.gov/). Ordinary DMSP-OLS night-time light has a problem of saturation in the bright cores of urban centres, with a maximum pixel value of 63, which makes it difficult to identify real hotspots. To solve this problem, NOAA/NGDC also publishes GRCNL with no sensor saturation. These images are produced by merging fixed-gain images, blending stable light as well as inter-satellite calibration (https://www.ngdc.noaa.gov/eog/dmsp/radcal_readme.txt). Therefore, it is comparable across different pixels. Empirical studies have shown that NTL can be used to map local economic activities [36–38], so that the spatial distribution of luminosity across the entire city can reveal the underlying urban form. As a result, urban hotspots can potentially be identified and extracted from NTL data.

## 2.3. Hotspot identification

A simple method to identify hotspots is to arbitrarily choose a threshold and any grid with its luminosity larger than this threshold would be considered a hotspot. However, setting a universal threshold could over- or under-estimate the numbers of hotspots due to the cities' heterogeneity in luminosity.

By contrast, Louail *et al.* [27] presented a method based on the Lorenz curve to generate a threshold for each city using cell phone traces. The quantile threshold can be calculated by the following formula:

$$F = 1 - \frac{\mu}{\rho_m}, \tag{2.1}$$

where $F$ is the quantile threshold, $\mu$ is the average density and $\rho_m$ is the maximum density. This criterion does not only depend on the average value of the density but also on the dispersion: as $\rho_m$ increases, the value of $F$ increases and therefore the number of identified hotspots decreases. Then, we can get the hotspot number $\mu/\rho_m$.

Furthermore, assuming that densities are ordered decreasingly $\rho_1 > \rho_2 > \ldots > \rho_n$, then, unit 1 is a complete hotspot, and based on the proportion to $\rho_1$, unit 2 is regarded as a $\rho_2/\rho_1$ hotspot. $\rho_n/\rho_1$ suggests unit $n$ has the probability of $\rho_n/\rho_1$ to be the same main hotspot as unit 1. Accordingly, unit $i$ adds $\rho_i/\rho_1$ to the total hotspot number. Therefore, we find the total hotspot number to be:

$$C_t = \sum_{n=1}^{N} \frac{\rho_i}{\rho_1}. \tag{2.2}$$

The proportion of hotspots is $C_t/N$, equal to $\mu/\rho_1$, which is the same as the result from the *Loubar* method.

This method offers another advantage in that it is not sensitive to city boundaries. Even if the statistical boundaries were to overstep the actual limits (covering much more non-constructive land in the outskirts), the result would be affected only minimally because the non-urbanized area features minimal light. This is especially relevant for the case of Chinese cities, where city boundaries can cover rural areas.

## 2.4. Characterizing the compactness of hotspots

Our hypothesis is that compactness of hotspots could be of significance to the economic performance of an urban environment. Following the work by Angel *et al.* [39], we develop two indexes to measure its compactness. Considering that a circle is the most compact shape, the indexes are based on comparing a metric of the set of hotspots with the same metric in a circle with the same area:

(1) Proximity index, PI, is the ratio between the diameter of a circle with the same area as the set of hotspots and the maximum distance between hotspots:

$$PI = \frac{D_d}{D_m}, \tag{2.3}$$

where PI is the proximity index, $D_d$ is the diameter of the circle of equal area to the set of hotspots and $D_m$ is the maximum distance between hotspots.

(2) Agglomeration index, AI, is the ratio between the average distance of hotspots to the geometric centre in a circle with the same area and the average distance to the corresponding geometric centre of the hotspots:

$$AI = \frac{D_e}{D_h}, \tag{2.4}$$

where AI is the agglomeration index, $D_e$ is the average distance of hotspot and the geometric centre of the circle of equal area that the set of hotspots and $D_h$ is the average distance of the hotspots to their corresponding geometric centre.

Both indexes quantify how hotspots sprawl over an urban region. The value of these two indicators range from 0 to 1; for cities with values close to 1, hotpots are close to each other (they are compact). By contrast, a value close to 0 indicates that hotspots are dispersed over the entire city. Consider the example presented in figure 2. Figure 2*a* (left) shows hotspots represented by red squares (raster units in the original NTL data). To measure its compactness, we compare it with the circle (figure 2*a* right) that has the same area as the actual set of hotspots. To calculate distances, we convert hotspots (squares) to points. In figure 2*a*, the maximum distance between hotspots is defined by the distance between points A and B: that is $D_m = D_{AB} = 10\sqrt{2}$. The diameter of the circle of equal area can be measured by the distance between points C and D: that is $D_d = D_{CD} = 10$. Then $PI = D_d/D_m = 0.71$.

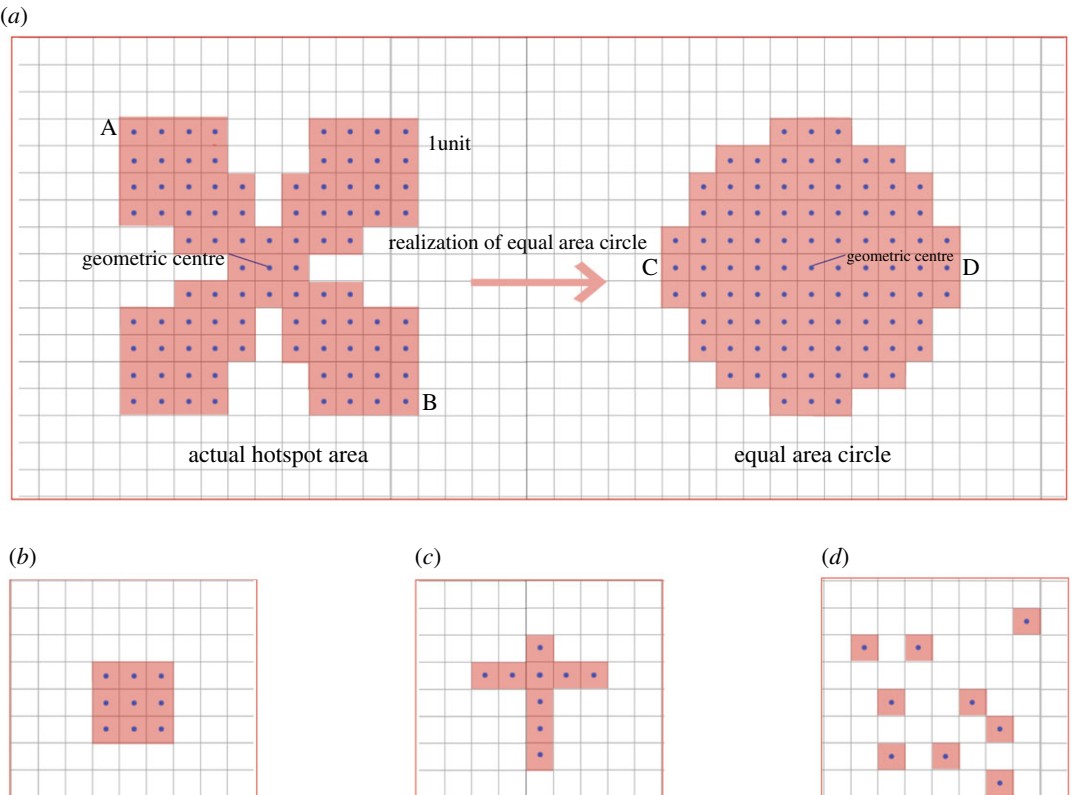

**Figure 2.** Example of a circle with the same area as the original set of hotspots and three examples representing high, middle and low compactness, respectively. (*a*) Hotspots are converted from raster units, the circle has the same area as the actual hotspot area: 89 units; (*b*) high compactness: both PI and AI approximate to 1; (*c*) middle compactness: PI is 0.68, while AI is 0.81; and (*d*) low compactness: PI is 0.4, while AI is 0.39.

The procedure of calculating AI is similar to that of PI. Three examples representing low, middle and high compact shapes are presented in figure 2*b*, *c* and *d*, respectively.

## 3. Results

First, we present the results regarding urban hotspot and population, which justify the use of NTL data. After that, we calculate the compactness index of every city considered, and visualize the relation between GDP per square kilometre (km²) and compactness across the USA, the EU and China. Finally, using least-squares, the inverted U-shaped effect of hotspot compactness on urban economic growth is presented.

### 3.1. Relation between number of hotspots and population

Louail *et al.* [27] demonstrated empirically that the number of hotspots scales sublinearly with the population size using Spanish anonymized and aggregated mobile phone data, which in turn corresponds to Louf & Barthelemy's theoretical work [18]. However, the paper does not show whether cities of other countries also follow such scaling law, especially for developing countries. In order to validate the use of NTL data for studying the structures of cities, we re-execute the experiment in Spanish cities. After that, we apply NTL data to American, European and Chinese cities, in order to find whether the sublinear relation is also valid.

Accordingly, the number of hotspots $N$ scales as

$$N = \alpha P^{\beta}, \tag{3.1}$$

where $N$ is the number of urban hotspots and $P$ is the population of the city.

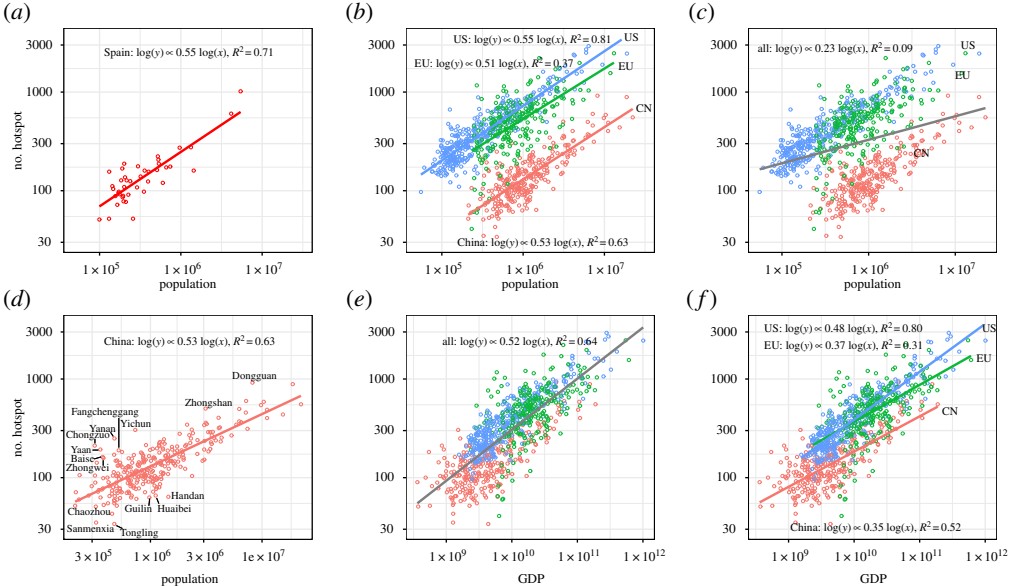

**Figure 3.** Sublinear relation using NTL data between the number of hotspots: (*a*) population for Spanish cities; (*b*) population for USA, EU and Chinese cities independently; (*c*) population and all three geographical areas; (*d*) population and Chinese cities for the study of outliers; (*e*) GDP for all three geographical areas and (*f*) GDP and each geographical area individually.

Figure 3*a* presents the scaling factor for Spanish cities using NTL. In this case, the scaling exponent is approximately 0.55, very similar to Louail *et al.'s* [27] result of 0.54. This result indicates that NTL data can be a relevant alternative for research on urban spatial structure, at least when compared with cell phone traces. Considering this result, we extend the NTL data to study the urban structure of other cities using hotspots. Figure 3*b* presents the scaling exponents for China, the EU and the USA with exponent values of 0.53, 0.50 and 0.55, respectively. These results reinforce the sublinear relation between the number of hotspots and city population for the three geographical areas considered.

Louf & Barthelemy [18] explain the appearance of subcentres as an effect of traffic congestion, and the number of subcentres scales sublinearly with the population size with a factor of $\mu/(\mu+1)$, where $\mu$ is the resilience of the transportation network to congestion. Our findings are similar to this theory; however, compared with its empirical work that shows a scaling exponent of 0.64, ours is smaller, ranging from 0.50 to 0.55. This difference may result from the definition of subcentre (hotspot) and grid value. Specifically, Louf & Barthelemy use a threshold method to identify subcentres, while this paper uses an endogenous method (*Loubar* method). The literature also shows that patent numbers and total wages [40,41] scale superlinearly with city population and that, on the other hand, gasoline sales [40], $CO_2$ emissions [42] and street length [43] scale sublinearly as a function of population size. In our context, urban hotspots also follow the sublinear scaling law. This sublinear relation suggests that, with population increasing by 100%, hotspot number just needs to increase by 50–55%. That is to say, the larger a city is, the more people a hotspot can support on average.

Compared with the similarity in the scaling values, the intersect values of these geographical areas differ greatly from one another. As portrayed in figure 3*b*, given a specific population, like one million, the fitted values of hotspots numbers are different among these three areas. Cities in the USA need the highest number of hotspots (724), while cities in China need the least (130). China has a total population of 1.3 billion, which contributes to larger cities. Besides, owing to the strict regulation on land use, cities become more agglomerated, which leads to denser and fewer hotspots. In the USA, the more cities sprawl, the more hotspots they need. As shown in figure 3*c*, globally, for the three areas considered, population does not explain this variation in the number of hotspots ($R^2 = 9\%$).

One of the most relevant contributors to this phenomenon, we hypothesize, is the gap between the economic development of the geographical areas considered. China is a developing economy, with a GDP *per capita* of 4000 Euro in 2010, which is smaller than those in USA and Europe. With Chinese economy improving its GDP, cities might tend to expand with time. As Angel *et al.* [44] empirically demonstrated, cities in higher income countries tend to feature lower population densities. To validate our hypothesis, we correlate number of hotspots to GDP and compare the result with population. Figure 3*e* presents the same study but considering GDP for the three areas considered, and the results

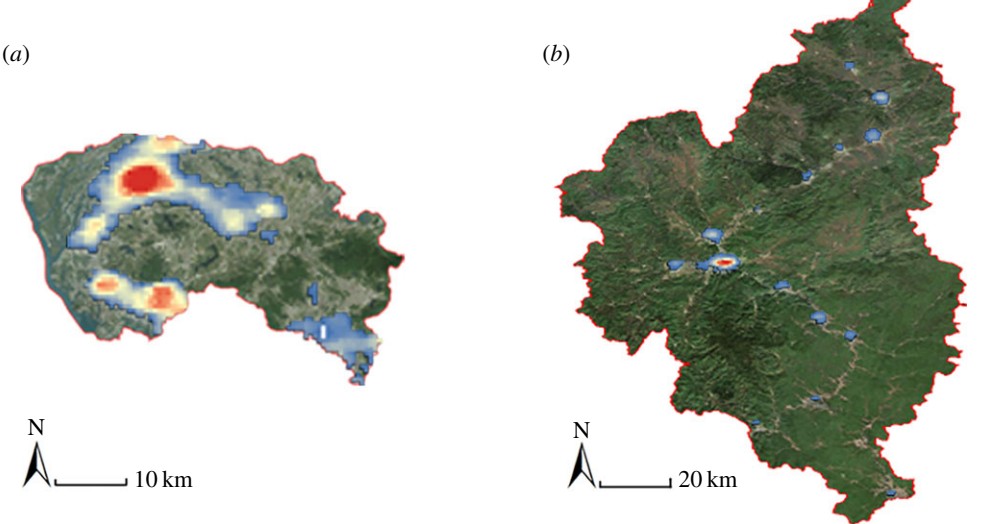

**Figure 4.** Two examples of outliers in China. (*a*) Dongguan. An outlier caused by development patterns, whose towns develop alone; and (*b*) Yichun, a valley city lying in a long narrow valley with many subcentres.

seem to indicate ($R^2 = 64\%$) that GDP can explain the majority of the variation in the number of hotspots. The same can be said when each geographical area is considered individually in figure 3*f* with values for $R^2$ of 80%, 52%, 32% for the USA, China and the EU, respectively. Our results indicate that GDP, rather than population size, has a global predictive power for the generation of hotspots.

Figure 3*f* also presents a set of relevant outlier cities, whose numbers of hotspots are unusually larger. Focusing on Chinese cities, in order to identify outliers, we calculate and normalize the residuals for each city in the regression analysis and classify cities as outliers if the residuals lie out of [−2,2]. As a result, we identified 15 outliers among 276 of China's cities, which are presented in figure 3*f*. We focus on two examples, Dongguan and Yichun, as examples to study why cities become outliers.

Dongguan was one of the first cities to open to foreign investment since China's reform policy was implemented. From that time on, towns in this city launched their urbanization process independently, even competing with each other. Specialized industrial towns stand out so much that no strong city core can be identified; thus, they constitute groups of towns rather than complete cities, and as a result their hotspots are over-represented for their population size (figure 4*a*). Yichun is a valley city, lying in a long and narrow valley, which makes it challenging to establish a strong centre. Instead, every district develops its own centre, which are then linearly connected and as a result, hotspots tend to be over-represented (figure 4*b*). This limited study indicates that outliers might arise primarily from development patterns or topography, as exemplified by Dongguan and Yichun, respectively.

## 3.2. Urban hotspot compactness

This section analyses how hotspots are organized spatially in urban areas. Figure 5 presents the frequency distribution of the PI and the AI for the cities in the USA, the EU and China. For the PI, the median value for China (figure 5*c*), the USA (figure 5*a*) and the EU (figure 5*b*) is 0.75, 0.49, and 0.43, respectively, while for the AI, the median values are 0.92, 0.78 and 0.69 for China (figure 5*f*), the USA (figure 5*d*) and the EU (figure 5*e*). These values indicate that hotspots tend to be more compact in Chinese cities followed by the USA and the EU.

This difference in both indexes may be the result of the urbanization gap between the geographical areas considered. Developed economies have reached a stable urbanization level with an urbanization rate over 75%, even experiencing suburbanization and counter-urbanization, which contributes to the dispersal of hotspots. However, developing economies, like China, are still in a rapid urbanizing process. The urbanization rate in China reached 57.4% in 2016. According to the S-shaped curve [45] used to depict the urbanization process, when the urbanization rate exceeds 70%, the urbanization development slows down and reaches its stage of mature development. Most developed economies have reached such mature stage while China is in its rapidly developing stage. China has a population of 1.3 billion, which means cities are supposed to absorb 0.3 billion more rural population

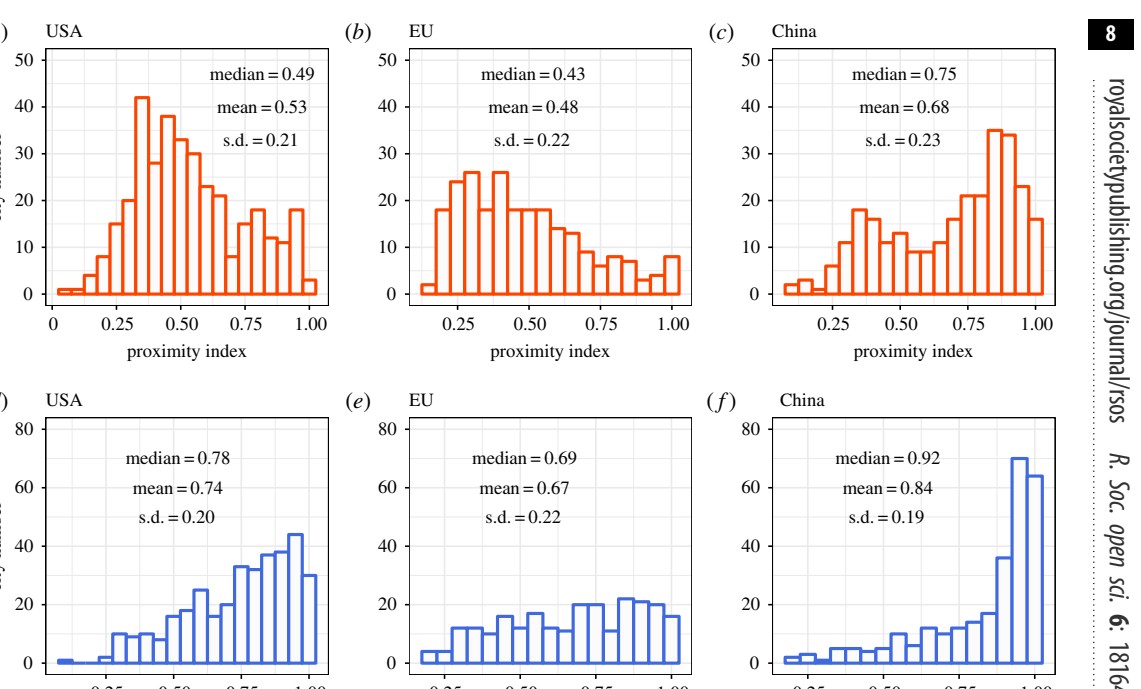

**Figure 5.** Frequency distribution of compactness in the USA, EU and China. (a,d) USA cities; both indexes show USA mean and median values are the second largest. (b,e) EU cities; both indexes show mean and median values are the smallest, which implies that in European cities hotspots are most disperse on average. (c,f) Chinese cities; both indexes show mean and median values are the largest, which means that hotspots in Chinese cities are on average closer to one another.

in the future when it reaches the urbanization rate of 70%. During the rapidly developing stage, the force of centralization is much stronger than that of decentralization. Specifically, urban service sectors, such as education, health care, retail and so on, usually appear in the inner city, which may force the hotspots to be more compact.

## 3.3. Non-monotonic effect of hotspot compactness on urban economic growth

Our hypothesis is that hotspot compactness can be of significance to economic performance. In order to validate it, we correlate compact indexes to GDP per $km^2$ and evaluate the effects of compactness on economic performance across the three geographical zones. Figure 6 presents the relation between PI and AI with log (GDP per $km^2$) for the USA (figure 6a,d); China (figure 6b,e) and the EU (figure 6c,f). The dominant relation in the majority of cases is an inverted U-shaped curve. That is especially the case the USA, where the inverted U-shaped relationship is quite strong ($R^2 = 0.16$ for PI and $R^2 = 0.13$ for AI). As for China, the inverted U-shaped effect is also visible, with $R^2 = 0.02$ for both proximity and AI. In the case of the EU, the U-shaped effect is weaker for the PI and not present for the AI.

In order to model the influence of hotspot compactness on urban economic growth we use a multiple regression analysis. Urban spatial structure transforms relatively slowly over time. Accordingly, we assume that the number of hotspots and their average spacing is exogenous to economic growth. We add population as a control variable in order to estimate the significance of the inverted U-shaped effect. The global model is as follows:

$$\ln Y = \beta_1 + \beta_2 \ln \text{Pop} + \beta_3 \text{Com} + \beta_4 \text{Com}^2 + \epsilon, \tag{3.2}$$

where $Y$ is the urban GDP per $km^2$, Pop is the population size and Com is the compactness index.

We focus our study in the three areas considered and for each one of them obtain five variations of the global regression model previously presented. Model 1 only considers population; Model 2 considers population plus the PI; Model 3 adds to Model 2 a quadratic term to the PI to check the relevance of the inverted U-shaped effect; Model 4 considers population plus the AI and Model 5 adds the quadratic element of the AI to Model 4.

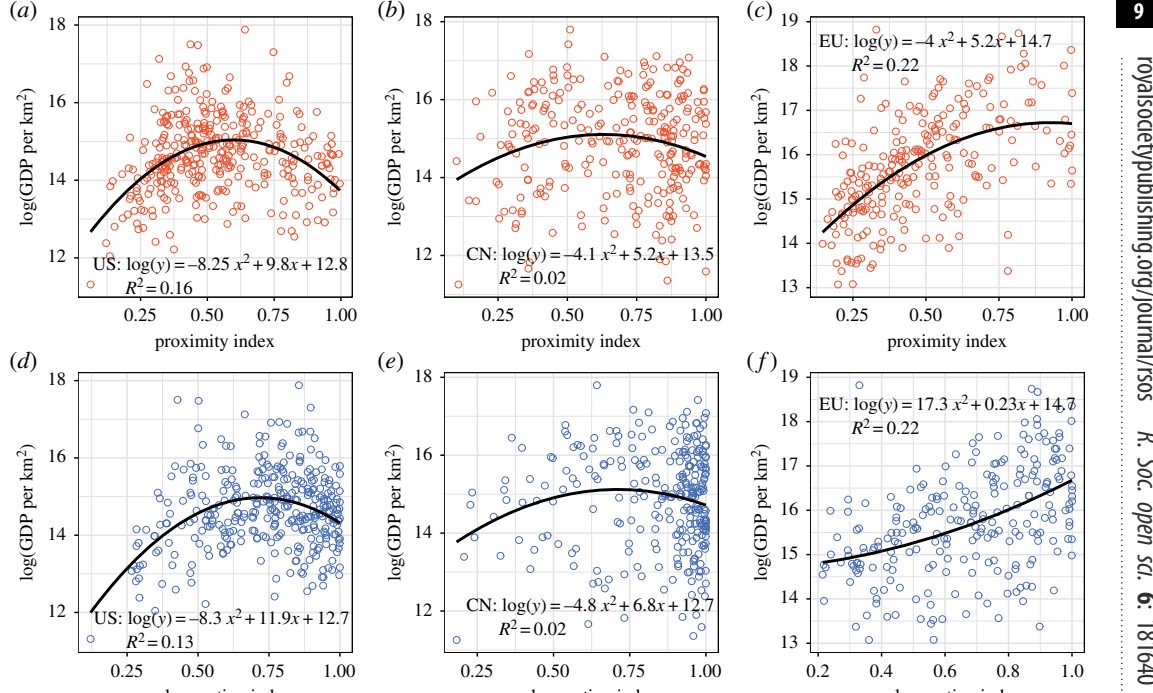

**Figure 6.** The inverted U-shaped curves for log (GDP per km$^2$) and compact indexes. (*a,d*) USA cities; both compactness indexes indicate a significant inverted U-shaped effect in the USA city system. (*b,e*) Chinese cities; the inverted U-shaped curves is visible but the effect is not significant. (*c,f*) EU cities; PI shows a stronger inverted U-shaped effect than AI.

**Table 1.** Regression results for the USA.

| | dependent variable: GDP per km$^2$ (log) | | | | |
| --- | --- | --- | --- | --- | --- |
| | (1) | (2) | (3) | (4) | (5) |
| population (log) | 0.652*** | 0.657*** | 0.599*** | 0.646*** | 0.605*** |
| PI | | 0.497*** | 6.340*** | | |
| PI$^2$ | | | −5.068*** | | |
| AI | | | | 0.549*** | 7.618*** |
| AI$^2$ | | | | | −5.235*** |
| constant | 6.372*** | 6.048*** | 5.329*** | 6.046*** | 4.407*** |
| observations | 349 | 349 | 349 | 349 | 349 |
| $R^2$ | 0.461 | 0.472 | 0.528 | 0.472 | 0.516 |
| AIC | 783.21 | 778.13 | 741.04 | 777.58 | 749.36 |
| F statistic | 296.534*** | 154.416*** | 128.440*** | 154.929*** | 122.705*** |

*$p < 0.1$; **$p < 0.05$; ***$p < 0.01$.

Table 1 presents the USA's regression results for the five variations of the global model, one per column. The results of Model 1 indicate that population exerts a significantly positive influence on the economic growth, while Model 2 shows that the regressive coefficient of the PI is significantly positive, which indicates that cities that are more compact tend to have higher GDP per km$^2$. As for Model 3, the quadratic term is significantly negative, and both coefficients of the PI and its quadratic term are statistically significant, which suggests that there exists an inverted U-shaped influence. As expected for Model 4, the regressive coefficient for the AI is also positive, highlighting again its influence on economic growth. In Model 5, the coefficient of the quadratic term is significantly negative, while the coefficient of AI is significantly positive, which again suggests the inverted U-shaped influence. Model 3 has the smallest AIC (741.04) and biggest $R^2$ (0.528), which indicates that

**Table 2.** Regression results for China.

| | dependent variable: GDP per km$^2$ (log) | | | | |
| | (1) | (2) | (3) | (4) | (5) |
|---|---|---|---|---|---|
| population (log) | 0.777*** | 0.800*** | 0.783*** | 0.792*** | 0.775*** |
| PI | | 0.520* | 3.854** | | |
| PI$^2$ | | | −2.696* | | |
| AI | | | | 0.547 | 3.992* |
| AI$^2$ | | | | | −2.478 |
| constant | 4.078*** | 3.413*** | 2.775** | 3.411*** | 2.598** |
| observations | 276 | 276 | 276 | 276 | 276 |
| $R^2$ | 0.257 | 0.266 | 0.276 | 0.263 | 0.269 |
| AIC | 843.9284 | 842.6548 | 840.8152 | 843.474 | 843.2493 |
| *F* statistic | 94.658*** | 49.348*** | 34.507*** | 48.797*** | 33.408*** |

*$p < 0.1$; **$p < 0.05$; ***$p < 0.01$.

**Table 3.** Regression results for the EU.

| | dependent variable: GDP per km$^2$ (log) | | | | |
| | (1) | (2) | (3) | (4) | (5) |
|---|---|---|---|---|---|
| population (log) | 0.460*** | 0.434*** | 0.438*** | 0.328*** | 0.319*** |
| PI | | 2.960*** | 7.501*** | | |
| PI$^2$ | | | −4.084*** | | |
| AI | | | | 2.251*** | 0.973 |
| AI$^2$ | | | | | 1.004 |
| constant | 9.477*** | 8.400*** | 7.303*** | 9.760*** | 10.238*** |
| observations | 239 | 239 | 239 | 239 | 239 |
| $R^2$ | 0.083 | 0.373 | 0.403 | 0.256 | 0.258 |
| AIC | 744.8238 | 656.0281 | 646.2218 | 696.6809 | 698.2188 |
| *F* statistic | 21.348*** | 70.073*** | 52.840*** | 40.655*** | 27.193*** |

*$p < 0.1$; **$p < 0.05$; ***$p < 0.01$.

is the best model that reveals the inverted U-shaped effect. Accordingly, when holding the other variables unchanged, a larger compactness coefficient corresponds to hotspots that are more compact and a greater advantage in terms of economic growth. However, after the compactness reaches a certain high level, its marginal effect on urban economic growth would become negative, suggesting an optimal compactness does exist. The optimal compactness coefficients of PI and AI are, respectively, 0.63 and 0.73.

When it comes to China, the inverted U-shaped effect is not as relevant but still statistically significant as showed in table 2. In this case, Model 3 also has the smallest AIC but its $R^2$ only increases 1% from Model 2. In the case of China, the quadratic term of the AI of Model 5 is not statistically significant. The results for the EU are presented in table 3 and similar to the previous case. Model 3 has the smallest AIC and in Model 5, the AI does not have a significant influence on GDP per km$^2$. These results also indicate that the PI may be more relevant to measure spatial compactness.

The results obtained in the USA, China and, to a lesser extent, in the EU, seem to validate the inverted U-shaped effect on economic growth of hotspot compactness. As a result, an optimal compactness for economic growth does exist that imposes the maximum positive influence on economic growth. Excessively close distances can bring congestion, but excessively large distances hinder connectivity. Therefore, hotspot spatial structure should be suitably compact.

# 4. Conclusion and discussion

Previous literature on the concept of compact city has focused mainly on the morphology of the urban environment. In this paper, we have presented a study of the effect of urban structural elements (hotspots) on the economic performance of the city. For that purpose, we used NTL data, which has the advantage of being globally available when compared with other sources such as cell phone traces or census data. Using the *Loubar* method, we identified and extracted hotspots of 864 cities in China, the EU and the USA. By comparison with the work by Louail *et al*. [27], that identified hotspots using cell phone traces in Spanish cities, we showed that NTL is an alternative data source for identifying hotspots.

The paper shows that the relationship between the number of hotspots and urban population follows a sublinear scaling law. With increasing population, hotspots scale sublinearly, with exponents of 0.50–0.55. Although China, the EU and the USA differ in population size, urbanization stage and economic development level, the exponent is nearly the same in the three cases. This result also supports the finding that urban spatial structure tends to be more polycentric with increasing population. We also found that the intercept values differ greatly among these three zones, which suggests that population by itself cannot explain the variation in the number of hotspots. Instead, urban economic size, represented by GDP, could be a more explanatory element for predicting the generation of hotspots.

The regression analysis indicated a U-shaped effect of hotspots spatial compactness on urban economic development; nevertheless, its visualization showed heterogeneous results. Specifically, the inverted U-shaped effect held for USA and EU cities, but for Chinese cities, the effect was not as clear. That can be explained mainly by two reasons: (i) in China, city boundaries are administrative rather than economic. These boundaries include areas that might not necessarily have strong economic interactions with the core urban areas, leading to a lower level of the GDP per $km^2$ and reducing the inverted U-shaped effect. (ii) AI might not be suitable for measuring compactness in this context, because it could overstate the hotspots spatial compactness. For future work, it would be relevant to construct city boundaries based on commute patterns to revisit the spatial compactness effect on urban economic growth in China. Also, alternative compactness measures could be constructed to test the robustness of the inverted U-shaped effect.

Data accessibility. Data available from the Dryad Digital Repository: https://doi.org/10.5061/dryad.2dt2d96 [46].

Authors' contributions. X.L., H.C. and W.X. designed the study. W.X. collected and analysed the data. X.L., E.F.-M., M.C., H.C. and W.X. interpreted the result and wrote the manuscript. All authors gave final approval for publication.

Competing interests. The authors declare no competing interests.

Funding. This study was supported by National Natural Science Foundation of China (grant no. 41571118).

Acknowledgements. We are grateful to three anonymous reviewers, who provided comments that substantially improved the manuscript. We also thanks editor(s) for the time to consider the manuscript.

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
