## [Reviewer comments · Royal Society Open Science]

Review History

RSOS-181640.R0 (Original submission)

Review form: Reviewer 1 (Horacio Samaniego)

Is the manuscript scientifically sound in its present form?

No

Are the interpretations and conclusions justified by the results?

No

Is the language acceptable?

No

Is it clear how to access all supporting data?

No

Do you have any ethical concerns with this paper?

No

Have you any concerns about statistical analyses in this paper?

Yes

Recommendation?

Major revision is needed (please make suggestions in comments)

Comments to the Author(s)

As in my first review of this ms, I find the topic addressed here very interesting, but have a hard time gauging whether this is really a novel piece of research. The potential is large, the topic timely, but the methods employed to address whether an “inverted U” shaped relationship exists between productivity and urban hotspots exists is tenuous to me. The ms needs some organization in terms of how the methods and results are presented. The fact that, as readers, we learn in the Results sections that a multiple linear regression is used to convey the idea that there is an optimal productivity associated to urban structure is just not right. I would hope that the authors could offer some more analysis in terms of model evaluation (In addition to use better statistical approximations to evaluate model likelihood such as AIC). Where does the square term comes from? I have no a priori problem with it, however, it would be nice to guide the reader and propose alternative hypotheses to be evaluated. Could you not use stepwise approaches towards such aim? As it is, I find the ms a bit deceiving.

Small concerns:

p2 | l3 ...researchers found *that* polycentric...

Papers to be considered in the discussion: Rozenfeld et al PNAS 2008 en sequels on mapping urban density and agglomerations

p2 | l29 I would rephrase the sentence starting with Theoretically, ... and seek another wording for the second avoid...

the term “suitable distance” is vague. Please be more specific. What is more suitable?

Materials and Methods

The novelty of using NTL data seems very promising. However, could we include a discussion on the caveats of NTL Data for urban economics research? What about sensor’s saturation? How does that affect the samples you have taken?

p3 | l18 Insert space between “Statistics (NUT)”

Eq 2.2 & 2.3 I re-read the ms a couple time to understand what the actual data is and what the qualitative differences are between both indices. What are you trying to emphasize by using a second index. Both provide very similar results.

p4 | l50 The universality of urban scaling is now hotly contested by several lines of research. Depersin & Barthelemy 2018, Arcaute et al 2014, Leitao et al 2016 among others... all point towards stating that while scaling seems to be important and prevalent among urban systems, the different regimes of urban scaling might be much harder to identify. I would remove the sentence starting with Actually, ... or explain it much more!! And there is the work o Lobo and Bettencourt on productivity (PloS One) showing how productivity scale with city size. What is the relation of those finding and yours? I miss that discussion or even the acknowledgment of such.

p4 | l58 remove “more”. It should read: The larger the XX, the more people...

Figure 2. A fair amount of scatter is seen for EU. Why? It seems more that some sort of constraint may be limiting the number of hotspots on the larger quantiles. I see an upward triangle in the data cloud. Why is that? Also, it would be nicer to keep the same axes for all four plots, so that the reader can gauge and compare the datasets.

p5 | 147 remove a dot at the end of line.

| 57 there's a space after outliers and comma.

p6 | end of first paragraph, this is more of a qualitative explanation, I do not see how the data reflects this...

Fig 3. can you pinpoint these 2 cities in the scaling plot? How much of an outlier are they? Could we identify other outliers that are not discussed here?

p6 | 137 please revise english of this sentence.

| 143 avoid repeats

| 155 we either need a citation or an explanation as to under which model are we expecting cities in CN to absorb 0.3 blns of rural pop.

| 157 I would move this to the discussion.

p7 | 135 insert comma after economic growth

| 153 move dot after citations

p8 | 115 we usually say: statistically significant (not the other way around)

| 120 This is the highlight of this research, yet, we need to show that this is the most likely situation among other possible models. I would also suggest using Akaike's Information Criterion to evaluate between them and even using a stepwise model selection.

Table 1. We only need one R^2 , possible the adj.

p9 | 17 Use past tense: we have present*ed*

| 18 insert a comma after: the Loubar method,

Review form: Reviewer 2

Is the manuscript scientifically sound in its present form?

Yes

Are the interpretations and conclusions justified by the results?

Yes

Is the language acceptable?

Yes

Is it clear how to access all supporting data?

Yes

Do you have any ethical concerns with this paper?

No

Have you any concerns about statistical analyses in this paper?

No

Recommendation?

Major revision is needed (please make suggestions in comments)

Comments to the Author(s)

This manuscript presents an hotspot analysis based on nighttime luminosity in several cities in US, EU and China. I did not review the first submission of this manuscript, but I assessed the present version of the manuscript, as well as the response to the original reviewer comments. My overall impression is positive; the topic is quite interesting and the paper is well-written. I therefore believe that it has the potential to be published in Royal Society Open Science, but there are several important points that must be addressed before publication.

- 1) I strongly encourage the authors to add a "related work" section to relate more deeply their work to the recent literature and clearly highlight the contribution to it.
- 2) I recommend the authors to add a map displaying the city location.
- 3) It is probably not the case but the fact that the granularity is expressed in arcsecond might have an effect on the results according to the latitude coordinate of the city. The authors could have made the choice of reprojecting and reaggregating the original raster data. I recommend the authors to add a sentence justifying this choice in the main text.
- 4) I do not understand the meaning of "diameter of equal area circle". More details are needed to help the readers better understand how PI and AI have been computed.
- 5) More discussion are needed regarding the regression model and the inverted U-shaped. It is not clear to me what was the criteria used to build the model and choose the variables and their shape (log and \wedge). The impact of the U shape is not clear either, it could be interesting to have some figures showing, for example, how an increase of x% of the hotspot compactness impacts the GDP.

Minor comments

- Line 13 Page 2: how the compactness of hotspots affects
- Line 47 Page 5: remove a period after hotspot
- Page 7: Replace "dummy variables" by "control variables"
- Last line Page 7: trough → through
- Page 9 Line 22: demonstrate → demonstrate

Review form: Reviewer 3

Is the manuscript scientifically sound in its present form?

Yes

Are the interpretations and conclusions justified by the results?

Yes

Is the language acceptable?

Yes

Is it clear how to access all supporting data?

Yes

Do you have any ethical concerns with this paper?

No

Have you any concerns about statistical analyses in this paper?

No

Recommendation?

Accept with minor revision (please list in comments)

Comments to the Author(s)

I enjoyed reading the paper. I think it should be published in RSOS, but after authors make some changes. The paper can be much more than it is currently. Honestly, I was put off by both the language in the response to the reviewers and also in the paper. It is too informal, too imprecise and tongue in cheek.

Authors took night time luminescence data from satellites and used it as a proxy of economic output. So far so good, although differences between countries should be noted. For example, US cares much more about light pollution than other cities might, so it may work to reduce it. To me, the most interesting result is the scaling law between population and the number of hotspots (city centers?). But here authors are so careless describing what they found and fail to impress the significance of the result. Here are two examples from the same paragraph

"Actually, scaling law is universal."

"On the other hand, the scaling of output, like Gasoline sales [39], CO2 emissions [41], and street length [42] are sublinear as a function of population size"

Actually, there is controversy about whether scaling law is universal (see <https://www.pnas.org/content/pnas/early/2018/02/16/1718690115.full.pdf>). But, leaving the controversy aside, how does the 0.5 slope relate to the city scaling body of work? Is it due to physical constraints? What are the gasoline or street length scaling exponents? This would be a great place to link the findings of this paper to that literature.

Another example of sloppiness is in the second sentence above - why is gasoline capitalized? Why is there a space before comma? Though hopefully this will be caught in the galley proofs, it is a good idea to carefully proof read and avoid such sloppy mistakes.

Decision letter (RSOS-181640.R0)

21-Feb-2019

Dear Dr Xu,

The editors assigned to your paper ("The inverted U-shaped effect of urban hotspots spatial compactness on urban economic growth") have now received comments from reviewers. We would like you to revise your paper in accordance with the referee and Associate Editor suggestions which can be found below (not including confidential reports to the Editor). Please note this decision does not guarantee eventual acceptance.

Please submit a copy of your revised paper before 16-Mar-2019. Please note that the revision deadline will expire at 00.00am on this date. If we do not hear from you within this time then it will be assumed that the paper has been withdrawn. In exceptional circumstances, extensions may be possible if agreed with the Editorial Office in advance. We do not allow multiple rounds of revision so we urge you to make every effort to fully address all of the comments at this stage. If deemed necessary by the Editors, your manuscript will be sent back to one or more of the original reviewers for assessment. If the original reviewers are not available, we may invite new reviewers.

- Data accessibility

If you wish to submit your supporting data or code to Dryad (<http://datadryad.org/>), or modify your current submission to dryad, please use the following link:
<http://datadryad.org/submit?journalID=RSOS&manu=RSOS-181640>

- Competing interests

- Authors' contributions

- Acknowledgements

- Funding statement

Kind regards,

Andrew Dunn

on behalf of Prof Jon Blundy (Subject Editor)

Comments to Author:

Reviewers' Comments to Author:

Reviewer: 1

Comments to the Author(s)

As in my first review of this ms, I find the topic addressed here very interesting, but have a hard time gauging whether this is really a novel piece of research. The potential is large, the topic

timely, but the methods employed to address whether an “inverted U” shaped relationship exists between productivity and urban hotspots exists is tenuous to me. The ms needs some organization in terms of how the methods and results are presented. The fact that, as readers, we learn in the Results sections that a multiple linear regression is used to convey the idea that there is an optimal productivity associated to urban structure is just not right. I would hope that the authors could offer some more analysis in terms of model evaluation (In addition to use better statistical approximations to evaluate model likelihood such as AIC). Where does the square term comes from? I have no a priori problem with it, however, it would be nice to guide the reader and propose alternative hypotheses to be evaluated. Could you not use stepwise approaches towards such aim? As it is, I find the ms a bit deceiving.

Small concerns:

p2 | 13 ...researchers found *that* polycentric...

Papers to be considered in the discussion: Rozenfeld et al PNAS 2008 en sequels on mapping urban density and agglomerations

p2 | 129 I would rephrase the sentence starting with Theoretically, ... and seek another wording for the second avoid...

the term “suitable distance” is vague. Please be more specific. What is more suitable?

Materials and Methods

The novelty of using NTL data seems very promising. However, could we include a discussion on the caveats of NTL Data for urban economics research? What about sensor’s saturation? How does that affect the samples you have taken?

p3 | 18 Insert space between “Statistics (NUT)”

Eq 2.2 & 2.3 I re-read the ms a couple time to understand what the actual data is and what the qualitative differences are between both indices. What are you trying to emphasize by using a second index. Both provide very similar results.

p4 | 150 The universality of urban scaling is now hotly contested by several lines of research. Depersin & Barthelemy 2018, Arcaute et al 2014, Leitao et al 2016 among others... all point towards stating that while scaling seems to be important and prevalent among urban systems, the different regimes of urban scaling might be much harder to identify. I would remove the sentence starting with Actually, ... or explain it much more!! And there is the work o Lobo and Bettencourt on productivity (PloS One) showing how productivity scale with city size. What is the relation of those finding and yours? I miss that discussion or even the acknowledgment of such.

p4 | 158 remove “more”. It should read: The larger the XX, the more people...

Figure 2. A fair amount of scatter is seen for EU. Why? It seems more that some sort of constraint may be limiting the number of hotspots on the larger quantiles. I see an upward triangle in the data cloud. Why is that? Also, it would be nicer to keep the same axes for all four plots, so that the reader can gauge and compare the datasets.

p5 | 147 remove a dot at the end of line.

| 57 there’s a space after outliers and comma.

p6 | end of first paragraph, this is more of a qualitative explanation, I do not see how the data reflects this...

Fig 3. can you pinpoint these 2 cities in the scaling plot? How much of an outlier are they?
Could we identify other outliers that are not discussed here?

p6 | 137 please revise english of this sentence.

| 143 avoid repeats

| 155 we either need a citation or an explanation as to under which model are we expecting cities in CN to absorb 0.3 blns of rural pop.

| 157 I would move this to the discussion.

p7 | 135 insert comma after economic growth

| 153 move dot after citations

p8 | 115 we usually say: statistically significant (not the other way around)

| 120 This is the highlight of this research, yet, we need to show that this is the most likely situation among other possible models. I would also suggest using Akaike's Information Criterion to evaluate between them and even using a stepwise model selection.

Table 1. We only need one R^2 , possible the adj.

p9 | 17 Use past tense: we have present*ed*

| 18 insert a comma after: the Loubar method,

Reviewer: 2

Comments to the Author(s)

This manuscript presents an hotspot analysis based on nighttime luminosity in several cities in US, EU and China. I did not review the first submission of this manuscript, but I assessed the present version of the manuscript, as well as the response to the original reviewer comments. My overall impression is positive; the topic is quite interesting and the paper is well-written. I therefore believe that it has the potential to be published in Royal Society Open Science, but there are several important points that must be addressed before publication.

- 1) I strongly encourage the authors to add a "related work" section to relate more deeply their work to the recent literature and clearly highlight the contribution to it.
- 2) I recommend the authors to add a map displaying the city location.
- 3) It is probably not the case but the fact that the granularity is expressed in arcsecond might have an effect on the results according to the latitude coordinate of the city. The authors could have made the choice of reprojecting and reaggregating the original raster data. I recommend the authors to add a sentence justifying this choice in the main text.

4) I do not understand the meaning of “diameter of equal area circle”. More details are needed to help the readers better understand how PI and AI have been computed.

5) More discussion are needed regarding the regression model and the inverted U-shaped. It is not clear to me what was the criteria used to build the model and choose the variables and their shape (log and \wedge). The impact of the U shape is not clear either, it could be interesting to have some figures showing, for example, how an increase of x% of the hotspot compactness impacts the GDP.

Minor comments

- Line 13 Page 2: how the compactness of hotspots affects
- Line 47 Page 5: remove a period after hotspot
- Page 7: Replace “dummy variables” by “control variables”
- Last line Page 7: trough → through
- Page 9 Line 22: demonstrate → demonstrate

Reviewer: 3

Comments to the Author(s)

I enjoyed reading the paper. I think it should be published in RSOS, but after authors make some changes. The paper can be much more than it is currently. Honestly, I was put off by both the language in the response to the reviewers and also in the paper. It is too informal, too imprecise and tongue in cheek.

Authors took night time luminescence data from satellites and used it as a proxy of economic output. So far so good, although differences between countries should be noted. For example, US cares much more about light pollution than other cities might, so it may work to reduce it. To me, the most interesting result is the scaling law between population and the number of hotspots (city centers?). But here authors are so careless describing what they found and fail to impress the significance of the result. Here are two examples from the same paragraph

"Actually, scaling law is universal."

"On the other hand, the scaling of output, like Gasoline sales [39], CO2 emissions [41], and street length [42] are sublinear as a function of population size"

Actually, there is controversy about whether scaling law is universal (see <https://www.pnas.org/content/pnas/early/2018/02/16/1718690115.full.pdf>). But, leaving the controversy aside, how does the 0.5 slope relate to the city scaling body of work? Is it due to physical constraints? What are the gasoline or street length scaling exponents? This would be a great place to link the findings of this paper to that literature.

Another example of sloppiness is in the second sentence above - why is gasoline capitalized? Why is there a space before comma? Though hopefully this will be caught in the galley proofs, it is a good idea to carefully proof read and avoid such sloppy mistakes.

Author's Response to Decision Letter for (RSOS-181640.R0)

See Appendix A.

RSOS-181640.R1 (Revision)

Review form: Reviewer 2

Is the manuscript scientifically sound in its present form?

Yes

Are the interpretations and conclusions justified by the results?

Yes

Is the language acceptable?

Yes

Is it clear how to access all supporting data?

Yes

Do you have any ethical concerns with this paper?

No

Have you any concerns about statistical analyses in this paper?

No

Recommendation?

Accept with minor revision (please list in comments)

Comments to the Author(s)

The authors have addressed most of my concerns. Nevertheless, I still have some reservations regarding the inverted U-shaped relationship. I am not completely convinced by the relationships (and associated R² values) displayed in Figure 6, even in the case of US for which, according to the authors, a “quite strong” relationship can be observed. This statement must be nuanced, I recommend the authors to add a small paragraph in the conclusion to discuss this relationship and the negative influence that seems generated by the excessive compactness.

- The caption is missing in Figure 6.

- page 9 line 33: I recommend the authors to change the term “urban performance” for “urban economic growth”

Review form: Reviewer 3

Is the manuscript scientifically sound in its present form?

Yes

Are the interpretations and conclusions justified by the results?

Yes

Is the language acceptable?

Yes

Is it clear how to access all supporting data?

Yes

Do you have any ethical concerns with this paper?

No

Have you any concerns about statistical analyses in this paper?

No

Recommendation?

Accept as is

Comments to the Author(s)

The authors have addressed my comments sufficiently. I have no objections to seeing this manuscript published.

Decision letter (RSOS-181640.R1)

23-Aug-2019

Dear Not specified Xu:

On behalf of the Editors, I am pleased to inform you that your Manuscript RSOS-181640.R1 entitled "The inverted U-shaped effect of urban hotspots spatial compactness on urban economic growth" has been accepted for publication in Royal Society Open Science subject to minor revision in accordance with the referee suggestions. Please find the referees' comments at the end of this email.

The reviewers and Subject Editor have recommended publication, but also suggest some minor revisions to your manuscript. Therefore, I invite you to respond to the comments and revise your manuscript.

Note that both previous reviewers have assessed the revisions made. Reviewer 2 now confirms that the manuscript is acceptable for publication, but Reviewer 1 is also mostly satisfied with the adjustments made, however they still have some reservations regarding the inverted U-shaped relationship; their concerns need to be addressed before we can accept your manuscript. I believe that these minor changes should not take long.

- **Ethics statement**

- **Data accessibility**

It is a condition of publication that all supporting data are made available either as supplementary information or preferably in a suitable permanent repository. The data accessibility section should state where the article's supporting data can be accessed. This section should also include details, where possible of where to access other relevant research materials

such as statistical tools, protocols, software etc can be accessed. If the data has been deposited in an external repository this section should list the database, accession number and link to the DOI for all data from the article that has been made publicly available. Data sets that have been deposited in an external repository and have a DOI should also be appropriately cited in the manuscript and included in the reference list.

If you wish to submit your supporting data or code to Dryad (<http://datadryad.org/>), or modify your current submission to dryad, please use the following link:
<http://datadryad.org/submit?journalID=RSOS&manu=RSOS-181640.R1>

- **Competing interests**

- **Authors' contributions**

- **Acknowledgements**

- **Funding statement**

Because the schedule for publication is very tight, it is a condition of publication that you submit the revised version of your manuscript before 01-Sep-2019. Please note that the revision deadline will expire at 00.00am on this date. If you do not think you will be able to meet this date please let me know immediately.

on behalf of Jon Blundy (Subject Editor)
openscience@royalsociety.org

Reviewer comments to Author:
Reviewer: 2

The authors have addressed most of my concerns. Nevertheless, I still have some reservations regarding the inverted U-shaped relationship. I am not completely convinced by the relationships (and associated R² values) displayed in Figure 6, even in the case of US for which, according to the authors, a "quite strong" relationship can be observed. This statement must be nuanced, I

recommend the authors to add a small paragraph in the conclusion to discuss this relationship and the negative influence that seems generated by the excessive compactness.

- The caption is missing in Figure 6.
- page 9 line 33: I recommend the authors to change the term “urban performance” for “urban economic growth”

Reviewer: 3

Comments to the Author(s)

The authors have addressed my comments sufficiently. I have no objections to seeing this manuscript published.

Author's Response to Decision Letter for (RSOS-181640.R1)

See Appendix B.

Decision letter (RSOS-181640.R2)

23-Oct-2019

Dear Mr Li,

I am pleased to inform you that your manuscript entitled "The inverted U-shaped effect of urban hotspots spatial compactness on urban economic growth" is now accepted for publication in Royal Society Open Science.

on behalf of Mr Andrew Dunn (Associate Editor) and Jon Blundy (Subject Editor)
openscience@royalsociety.org

Appendix A

Dear Andrew Dunn,

Thank you for the opportunity to submit our revised manuscript for your consideration in *Royal Society Open Science*. We reply to each of your and the reviewers' comments in sequence, with special attention to the regression models that prove the existence of U-shape effects of urban hotspots spatial structure on urban economic performance. In addition to the strengthened regression result, we also visualized the inverted U-shaped curve for all three economies. We would also like to specifically thank you and the reviewers for the detailed, helpful feedback on this manuscript.

Reviewer: 1

Q1:

As in my first review of this ms, I find the topic addressed here very interesting, but have a hard time gauging whether this is really a novel piece of research. The potential is large, the topic timely, but the methods employed to address whether an “inverted U” shaped relationship exists between productivity and urban hotspots exists is tenuous to me. The ms needs some organization in terms of how the methods and results are presented. The fact that, as readers, we learn in the Results sections that a multiple linear regression is used to convey the idea that there is an optimal productivity associated to urban structure is just not right. I would hope that the authors could offer some more analysis in terms of model evaluation (In addition to use better statistical approximations to evaluate model likelihood such as AIC). Where does the square term comes from? I have no a priori problem with it, however, it would be nice to guide the reader and propose alternative hypotheses to be evaluated. Could you not use stepwise approaches towards such aim? As it is, I find the ms a bit deceiving.

A1:

We thank the reviewer for the detailed comments and believe we address the primary concerns in our revised submission.

Here is our hypothesis regarding how hotspot spatial structured affect urban economic performance. We believe if hotspots are spaced over-dispersedly, the communication and transportation costs between people and goods could increase to a great extent. However, over-compactness would generate congestions that would make cities function less efficiently. Therefore, we hypothesize that there could exist an optimal hotspot compactness that help city achieve economic success. This hypothesis is further discussed in Section 3.2 now. To prove the hypothesis, we modified the manuscript in two aspects. Firstly, we visualize the relationships between GDP per km² and hotspot compactness index across the US, EU and China, as showed in figure 6. The inverted U-shape curve is most significant in US, followed by China. However, the inverted U-shape is weak in EU. Secondly, we also carry out regression analysis on each economy respectively, using population as a control variable to test the significance of hotspot compactness in the regression models. The models show the hypothesis is acceptable.

As you suggested, we adopt AIC method to evaluate the regression models. It turns out that the

models with the quadratic term of PI has the smallest AIC. And the coefficients of each term are also statistically significant. We appreciate your suggestion about the AIC method and that helps clarify the regression result.

The square term, or quadratic term is used to test if there exists a U shape or inverted U-shape. We added it to the basic model to test our hypothesis.

Finally, we think the stepwise approach is necessary, as it can show all the regression processes, and make different models comparable. It can show to what extent the square term can affect the dependent variable. Thanks for your comments.

Q2:

p2|13 ...researchers found *that* polycentric...

Papers to be considered in the discussion: Rozenfeld et al PNAS 2008 en sequels on mapping urban density and agglomerations

A2.

Actually, gridded data is useful for identify urban clusters and urban hotspots. Thanks for your recommendation.

Q3:

p2|129 I would rephrase the sentence starting with Theoretically, ...

and seek another wording for the second avoid...

the term "suitable distance" is vague. Please be more specific. What is more suitable?

A3:

Honestly, it is not a scientific statement. This time I have deleted it. Thanks!

Q4:

Materials and Methods

The novelty of using NTL data seems very promising. However, could we include a discussion on the caveats of NTL Data for urban economics research? What about sensor's saturation? How does that affect the samples you have taken?

A4:

Thanks for your comments re the saturation issue. In the M&M section of the revised manuscript, we discuss this saturation issue and address how we solve it. Ordinary DMSP-OLS nighttime light has saturation issue in the bright cores of the urban centers, with a maximum pixel value of 63, which makes it difficult to identify the real hotspots. To solve this problem, NOAA/NGDC published GRCNL with no sensor saturation. These images are produced by merging fixed-gain images, blending stable light as well as inter-satellite calibration. This study uses the GRCNL dataset.

Another problem is the distortion of map projection that would influence the estimation of urban structure and morphology. Specifically, different projection will influence the accuracy of

calculating distance between different hotspot. Based on the Arcgis 10.0, we change the projection of every city to its corresponding Gauss-Kruger projection. The standard zones of Gauss-Kruger projection are three degree wide, and cities are distributed to the standard zones according to their central meridian.

Q5

p3|18 Insert space between “Statistics (NUT)”

A5

We have revised it. Thanks.

Q6:

Eq 2.2 & 2.3 I re-read the ms a couple time to understand what the actual data is and what the qualitative differences are between both indices. What are you trying to emphasize by using a second index. Both provide very similar results.

A6

Actually, there are many methods to measure the compactness. Here we use two indexes in order to make the result more robust. Both measures lead to the same result, that is, the quadratic terms are statistically significant and negative values, proving our hypothesis.

Q7

p4|150 The universality of urban scaling is now hotly contested by several lines of research. Depersin & Barthelemy 2018, Arcaute et al 2014, Leitao et al 2016 among others... all point towards stating that while scaling seems to be important and prevalent among urban systems, the different regimes of urban scaling might be much harder to identify. I would remove the sentence starting with Actually, ... or explain it much more!! And there is the work o Lobo and Bettencourt on productivity (PloS One) showing how productivity scale with city size. What is the relation of those finding and yours? I miss that discussion or even the acknowledgment of such.

A7

The papers you suggested give us a bigger picture. Although scaling is prevalent among urban systems, many other factors, not just population, might contribute to the urban scaling. For example, in the work by Depersin & Barthelemy, the congestion-induced delay in a given city does not depend on its population only, but also on its history. In the revised manuscript, we relate the hotspots to GDP, and it turns out that GDP have more predictive powers for the development of hotspots. The sentence - “actually, scaling laws is universal”, has been deleted accordingly.

In the work by Lobo and Bettencourt, the productivity scales super-linearly to the city size. The productivity is represented by the urban total wage. Using our data, we find the scaling exponents are 1.21, 1.07 and 1.10 for China, EU and the US respectively in terms of GDP. However, this is beyond the focus of this paper, so we haven't incorporated them.

Q8

p4|158 remove “more”. It should read: The larger the XX, the more people...

A8

Revised accordingly. Thank you.

Q9

Figure 2. A fair amount of scatter is seen for EU. Why? It seems more that some sort of constraint may be limiting the number of hotspots on the larger quantiles. I see an upward triangle in the data cloud. Why is that? Also, it would be nicer to keep the same axes for all four plots, so that the reader can gauge and compare the datasets.

A9

We have revised figure 2 (now figure 3), putting different economies' scatters in one plot, so that we can make comparisons across them. The R^2 of EU is quite small, only 37%. And we see an upward triangle about it. We think that might be caused by different definition of city boundary across three economies.

Q10

p5|47 remove a dot at the end of line.

|57 there's a space after outliers and comma.

A10:

Revised accordingly. Thank you.

Q11

p6| end of first paragraph, this is more of a qualitative explanation, I do not see how the data reflects this...

A11

The sentence has been changed to: outliers might arise primarily from development patterns or topography, as exemplified by Dongguan and Yichun respectively.

Q12

Fig 3. can you pinpoint these 2 cities in the scaling plot? How much of an outlier are they? Could we identify other outliers that are not discussed here?

A12:

We pinpoint these 2 cities as well as other cities in the scatter plot of Fig 3d. We get 15 outliers using the following method. We calculate and normalize the residuals for each city in the regression analysis of hotspot and population. Then we denote the cities as outliers if the residuals lie out of $[-2,2]$. Here we can get 15 outliers among 276 China's cities. They are shown in figure3d.

Q13

p6|37 please revise english of this sentence.

|I43 avoid repeats

A13

Revised accordingly. Thank you.

A14

|I55 we either need a citation or an explanation as to under which model are we expecting cities in CN to absorb 0.3 blns of rural pop.

A14

In Section 3.2 of the revised manuscript, we added the discussion about Ray.M Northam's S-shaped Curve. According to S-shaped curve, when the urbanization rate exceeds 70%, the urbanization development is going slower and reach its late stage of mature development. China is supposed to reach such a stage in the future.

Q15

|I57 I would move this to the discussion.

A15:

With the introduction of Ray.M Northam's S-shaped Curve before this sentence, we believe keeping it would make the manuscript easier to follow.

Q16

p7|I35 insert comma after economic growth

|I53 move dot after citations

p8|I15 we usually say: statistically significant (not the other way around)

A16:

Revised accordingly. Thank you.

Q17

|I20 This is the highlight of this research, yet, we need to show that this is the most likely situation among other possible models. I would also suggest using Akaike's Information Criterion to evaluate between them and even using a stepwise model selection.

A17.

See A1.

Q18

Table 1. We only need one R^2 , possible the adj.

p9|I7 Use past tense: we have present*ed*

|I8 insert a comma after: the Loubar method,

A18

Revised accordingly. Thank you.

Reviewer: 2

Comments to the Author(s)

This manuscript presents an hotspot analysis based on nighttime luminosity in several cities in US, EU and China. I did not review the first submission of this manuscript, but I assessed the present version of the manuscript, as well as the response to the original reviewer comments. My overall impression is positive; the topic is quite interesting and the paper is well-written. I therefore believe that it has the potential to be published in Royal Society Open Science, but there are several important points that must be addressed before publication.

We thank the reviewer for the detailed comments and believe we address the primary concerns in our revised submission.

Q1:

1) I strongly encourage the authors to add a “related work” section to relate more deeply their work to the recent literature and clearly highlight the contribution to it.

A1:

Thanks for your suggestion. Actually, we have incorporated related literature into the “introduction” section, which might make the manuscript more readable. We think our findings themselves could be the contribution to the recent literature. For example, we demonstrate the NTL data could be a good alternative for mobile phone data to measure urban structure. We find the scaling exponent is similar across different countries and an inverted U-shaped relationship exists between compactness and urban economic performance.

Q2

2) I recommend the authors to add a map displaying the city location.

A2

We have added the location map as Figure 1.

Q3

3) It is probably not the case but the fact that the granularity is expressed in arcsecond might have an effect on the results according to the latitude coordinate of the city. The authors could have made the choice of reprojecting and reaggregating the original raster data. I recommend the authors to add a sentence justifying this choice in the main text.

A3.

Thanks for the comment. The distortion of map projection that would influence the estimation of urban structure and morphology. Specifically, different projection will influence the accuracy of calculating distance between different hotspot. Based on the Arcgis 10.0, we change the projection of every city to its corresponding Gauss-Kruger projection. The standard zones of Gauss-Kruger projection are three degree wide, and cities are distributed to the standard zones according to their central meridian. This process can help get more accurate distances. Besides, As the compactness is calculated compared to the equal area circle that is generated in the same zone or

same latitude coordinate. This could reduce the inaccuracy to a great extent. The manuscript has been modified accordingly.

Q4.

4) I do not understand the meaning of “diameter of equal area circle”. More details are needed to help the readers better understand how PI and AI have been computed.

A4.

We have added an example to illustrate the process in Figure 2. For example, the maximum hotspots distance of the actual hotspots can be measured by the distance between point A and B: that is $D_m = D_{AB} = 200^{0.5}$. The diameter of equal area circle can be measured by the distance between point C and D: that is $D_d = D_{CD} = 10$. Then $PI = D_d / D_m = 0.71$. The procedure of calculating AI is similar to that of PI.

Figure 1 An example of actual hotspot and its equal area circle. Hotspots are converted from raster units named pixels. The equal area circle owns the same area to the actual hotspot area. Specifically, both of them have an area of 89 units.

Q5

5) More discussion are needed regarding the regression model and the inverted U-shaped. It is not clear to me what was the criteria used to build the model and choose the variables and their shape (log and \wedge^2). The impact of the U shape is not clear either, it could be interesting to have some figures showing, for example, how an increase of x% of the hotspot compactness impacts the GDP.

A5.

We make visualizations of the inverted U-shape relations between hotspot compactness and urban economic performance in Figure 6. The regression model is set to test whether the inverted U-shaped effect is statistically significant. As we know, population is the main factor contributing to economic growth, so we add it as the control variable. Take the US cities as examples, with PI increase by 0.1 from X, the GDP per km² would increase by $(9.8-16.5x)/1000$.

Figure 2 the relationship between GDP per km2 and two compact indexes in the US.

A5

Minor comments

- Line 13 Page 2: how the compactness of hotspots affects
- Line 47 Page 5: remove a period after hotspot
- Page 7: Replace “dummy variables” by “control variables”
- Last line Page 7: trough → through
- Page 9 Line 22: demonstrate → demonstrate

Q6.

Revised accordingly. Thank you.

Reviewer: 3

Comments to the Author(s)

I enjoyed reading the paper. I think it should be published in RSOS, but after authors make some changes. The paper can be much more than it is currently. Honestly, I was put off by both the language in the response to the reviewers and also in the paper. It is too informal, too imprecise and tongue in cheek.

Authors took night time luminescence data from satellites and used it as a proxy of economic output. So far so good, although differences between countries should be noted. For example, US cares much more about light pollution than other cities might, so it may work to reduce it. To me, the most interesting result is the scaling law between population and the number of hotspots (city centers?). But here authors are so careless describing what they found and fail to impress the significance of the result. Here are two examples from the same paragraph

"Actually, scaling law is universal."

"On the other hand, the scaling of output, like Gasoline sales [39], CO2 emissions [41], and street length [42] are sublinear as a function of population size"

Actually, there is controversy about whether scaling law is universal (see <https://www.pnas.org/content/pnas/early/2018/02/16/1718690115.full.pdf>). But, leaving the controversy aside, how does the 0.5 slope relate to the city scaling body of work? Is it due to physical constraints? What are the gasoline or street length scaling exponents? This would be a great place to link the findings of this paper to that literature.

A1:

We thank the reviewer for the detailed comments and believe we address the primary concerns in our revised submission.

Regarding the scaling exponents, we also think it's one of the most exciting findings in this study, especially we find the exponents are close to each other, from 0.5 to 0.55. Louf, R. & M. Barthelemy's work¹ have discussed the sublinear relationship between activity centers and city population. They build up a dynamic model and find the scaling exponent is $\mu / (\mu + 1)$, where μ is the resilience of transportation network to congestion. Our findings coincide with this theory. We complemented the manuscript with discussions about the scaling exponents.

Compared to its empirical finding that shows the scaling exponent is appropriately 0.64, ours is much smaller, ranging from 0.50 to 0.55. Actually, we think 0.64 is a little overestimated. In Bettencout's model, it is thought theoretically that land area scale sublinearly with the population sizes at the exponent of 2/3, and empirically, the exponent is 0.63 for the urbanized areas of US

¹ Louf, R. & M. Barthelemy, Modeling the polycentric transition of cities. Physical review letters, 2013. 111(19): p. 198702

MSAs. Cities are often compared to living organisms with hierarchical organization consisting of cells, tissues and organs. Among mammals, brains size sublinearly scale with the body size. Here if we regard urban centers as urban brains, land as body, then we hypothesize that the urban center scale sublinearly with land area. Then we could conclude the scaling exponent between population and urban size is thought to be smaller than $2/3$. However, it is just an analogy. We can't guarantee the explanation is right. If we have a land area data, that might work. The 0.5 slope seems need greatly more effort to discuss. Therefore, we decide not to incorporate the discussion above in the main text.

Q2:

What are the gasoline or street length scaling exponents?

A2:

The scaling exponents of gasoline and street length is 0.79 and 0.86, respectively. That also shows a sublinear relationship between resources used and population. In the main text, we haven't showed the exact number. That's because we care more about the sublinear relation than the exact number.

Q3:

Another example of sloppiness is in the second sentence above - why is gasoline capitalized? Why is there a space before comma? Though hopefully this will be caught in the galley proofs, it is a good idea to carefully proof read and avoid such sloppy mistakes.

A3:

Sorry for my mistakes about the expression. They are changed accordingly. Thanks for your comments.

Appendix B

Dear Alice Power,

Thank you for considering our manuscript to publish in *Royal Society Open Science*. We reply to each of your and the reviewers' comments in sequence. This time we mainly add a small paragraph to discuss the inverted U-shaped effect of hotspot. Besides, we also update some figures. We would also like to specifically thank you and the reviewers for the detailed, helpful feedback on this manuscript.

Reviewer: 2

Q1:

The authors have addressed most of my concerns. Nevertheless, I still have some reservations regarding the inverted U-shaped relationship. I am not completely convinced by the relationships (and associated R2 values) displayed in Figure 6, even in the case of US for which, according to the authors, a “quite strong” relationship can be observed. This statement must be nuanced, I recommend the authors to add a small paragraph in the conclusion to discuss this relationship and the negative influence that seems generated by the excessive compactness.

A1:

Thank you for your advice. Actually, the inverted U-shaped effect need far more discussion. According to your suggestion, this time we add a small paragraph to discuss why the inverted U-shaped effect occurs. Specifically, we tune down the inverted U-shaped effect in this version. The main change is in the following paragraph. Thanks a lot.

The regression analysis indicated a U-shaped effect of hotspots spatial compactness on urban economic development, nevertheless its visualization showed heterogeneous results. Specifically, the inverted U-shaped effect held for US and EU cities, but for Chinese cities, the effect was not as clear. That can be explained mainly by two reasons: (1) in China, city boundaries are administrative rather than economic. This boundaries include areas that might not necessarily have strong economic interactions with the core urban areas, leading to a lower level of the GDP per km² and reducing the inverted U-shaped effect; and (2) Agglomeration Index might not be suitable for measuring compactness in this context, because it could overstate the hotspots spatial compactness. For future work, it would be relevant to construct city boundaries based on commute patterns to revisit the spatial compactness effect on urban economic growth in China. Also, alternative compactness measures could be constructed to test the robustness of the inverted U-shaped effect.

Q2:

- The caption is missing in Figure 6.
- page 9 line 33: I recommend the authors to change the term “urban performance”

for “urban economic growth”

A2:

Thanks for your reminder. We have added the caption in Figure 6. Besides, “urban performance” have been changed to “urban economic growth”. Thanks a lot.

Reviewer: 3

Q1:

The authors have addressed my comments sufficiently. I have no objections to seeing this manuscript published.

A1:

Thanks for your support. You do give us great advice to promote this manuscript. Thanks again!